# Optimized Mass Spectrometry Detection of Thyroid Hormones and Polar Metabolites in Rodent Cerebrospinal Fluid

**DOI:** 10.3390/metabo14020079

**Published:** 2024-01-23

**Authors:** Ryann M. Fame, Ilhan Ali, Maria K. Lehtinen, Naama Kanarek, Boryana Petrova

**Affiliations:** 1Department of Pathology, Boston Children’s Hospital, Boston, MA 02115, USA; 2Harvard Medical School, Boston, MA 02115, USA; 3Department of Neurosurgery, Stanford University, Stanford, CA 94305, USA; 4Broad Institute of Harvard and MIT, Cambridge, MA 02142, USA

**Keywords:** thyroid hormone, metabolomics, mass spectrometry method, reverse-phase chromatography, cerebrospinal fluid, rodent, development

## Abstract

Thyroid hormones (TH) are required for brain development and function. Cerebrospinal fluid (CSF), which bathes the brain and spinal cord, contains TH as free hormones or as bound to transthyretin (TTR). Tight TH level regulation in the central nervous system is essential for developmental gene expression, which governs neurogenesis, myelination, and synaptogenesis. This integrated function of TH highlights the importance of developing precise and reliable methods for assessing TH levels in CSF. We report an optimized liquid chromatography–mass spectrometry (LC-MS)-based method to measure TH in rodent CSF and serum, applicable to both fresh and frozen samples. Using this new method, we find distinct differences in CSF TH in pregnant dams vs. non-pregnant adults and in embryonic vs. adult CSF. Further, targeted LC-MS metabolic profiling uncovers distinct central carbon metabolism in the CSF of these populations. TH detection and metabolite profiling of related metabolic pathways open new avenues of rigorous research into CSF TH and will inform future studies on metabolic alterations in CSF during normal development.

## 1. Introduction

Thyroid hormones (TH) are essential for brain development and function [1,2,3,4,5,6,7,8]. TH play direct roles in neural precursor proliferation and differentiation in embryonic neurogenic areas [9] and in ex vivo embryonic stem cells [10], with maternally provided TH levels playing a critical role [11]. Structures that differentiate postnatally, like the cerebellum or myelinating glia, are also sensitive to TH levels [12,13] and depend on postnatal TH for maintenance. Hypothyroidism, iodide deficiency, and mutations in TH receptors all result in developmental brain abnormalities [13,14,15]. In sum, maintaining balanced TH levels within the central nervous system (CNS) is vital for developmental health. Therefore, measuring TH concentrations directly within the CNS, rather than in the peripheral system, could enhance the accuracy of monitoring these essential biological pathways.

Cerebrospinal fluid (CSF) is a biofluid with roles in CNS development and function, and its contents often reflect CNS states. CSF TH levels depend on multiple processes, including systemic TH levels, activities of iodothyronine deiodinases that convert the pro-hormone T4 to active T3, blood–brain barrier permeability, and transport at the choroid plexus. Choroid plexus tissues are three-dimensional structures of specialized cuboidal epithelial cells surrounding fenestrated capillaries and mesenchyme that extend into the four brain ventricles. The choroid plexus secretes CSF in adults and comprises the blood–CSF barrier by limiting transcellular transport and maintaining tight junctions at the apical surface of choroid plexus epithelial cells. Choroid plexus epithelial cells transport TH into CSF by the TH carrier protein transthyretin (TTR), which is the major transporter of TH, and through the transporters MCT8 and OAT1C1 [16,17,18]. TTR prevents the loss of TH from CSF into the bloodstream in hypothyroid animals [19]. TH levels cycle in a circadian manner in serum [20] and concurrently, CSF TH levels and TTR cycle with circadian time [21]. The dynamic and physiologically responsive nature of CSF TH levels motivate the need for improved detection methods for TH in this biofluid.

The thyroid gland produces two major forms of TH: the active hormone 3,5,3′-L-triiodothyronine (T3) and the inactive pro-hormone, 3,5,3′5′-L-tetraiodothyronine (T4, aka thyroxine). While only about 10% of the TH produced by the thyroid gland is T3, iodothyronine deiodinases (DIO1, DIO2, DIO3) fine-tune TH homeostasis to mediate the activation of T4 to T3 (through the activities of DIO1 and DIO2) and inactivation of T3 to reverse-T3 (through DIO3) [22]. DIO1 is mainly localized in the liver, kidneys, and thyroid. DIO3 is abundant in the placenta and uterus during pregnancy, as well as in neurons and skin [23]. DIO2 is most highly expressed in the brain (largely in astrocytes), pituitary, thyroid, skeletal muscle, and brown adipose tissue [24]. Therefore, DIO2 is responsible for most T4-to-T3 conversion in the CNS with additional contribution from DIO3. Together, the two enzymes have the ability to rapidly increase activity in hypothyroid states and maintain CNS thyroid status [25]. Peripheral iodothyronine deiodinases also contribute to TH homeostasis since peripheral TH can be transported to the CNS. This conversion to T3 is critical as T3 is the key active TH that interacts with nuclear TH receptors to direct gene expression pathways that are vital for brain development and function [23,26,27,28]. Therefore, individual assessment of both the active form (T3) and the pro-hormone (T4) is key to accurately represent bioavailable TH reservoirs.

In the past, radioimmunoassays were the standard for measuring TH levels, but these methods were constrained by their narrow sensitivity and dynamic range. Liquid chromatography–mass spectrometry (LC-MS) techniques offer a substantial enhancement, providing a broader dynamic range and heightened sensitivity. Such sensitivity is especially useful for the detection of TH, as they are typically found in trace amounts, ranging from 1.4 to 2.3 pg/mL in biological samples [29], and their levels fluctuate dynamically. Recent advancements in LC-MS technologies have further simplified sample preparation and provided accurate TH quantification across various tissues and biofluids [30,31,32,33,34,35,36]. The sensitivity provided by LC-MS is even more crucial when samples are limited, as with CSF. Further, determining TH levels in CSF is challenging because of their low levels in the fluid. Average CSF TH levels have been reported at 1.85 pg/L for T4 and 0.17 pg/mL for T3. These levels are about 1/40 and 1/10 of serum levels, respectively [37]. Few methods address these dual challenges of sample scarcity and low levels, leading to infrequent reports on CSF hormone levels [30]. A further advantage of using LC-MS to quantify TH in CSF is that it can be combined with global assessments of the metabolic state through targeted LC-MS metabolic profiling of central energy metabolism, a process dependent on TH during development and adulthood [8,23,27,38,39,40,41]. Better detection and reporting of this key hormone in CSF requires a new and sensitive approach, which we describe here.

In this study, we present a quantitative LC-MS method to detect T3 and T4 hormone levels in CSF from both fresh and frozen samples. The method is based on high-resolution accurate mass detection via Orbitrap MS and on separation of hormone forms via reverse-phase chromatography. We optimize several aspects of signal detection, explore diverse chromatographic methods, and improve sample preparation. We confirm lower levels of TH in CSF compared to serum and further report dynamic developmental and physiological differences. Finally, our method provides a significant advantage over previous methods as it combines TH measurements with a more comprehensive characterization of central energy metabolism that enables monitoring of downstream metabolic and signaling effects.

## 2. Materials and Methods

### 2.1. Animals

The Boston Children’s Hospital IACUC approved all experiments involving mice in this study. Adult CD-1 male and female mice and pregnant CD-1 dams were obtained from Charles River Laboratories. Sprague Dawley rats were purchased from Charles River Laboratories. Animals were housed in a temperature- and humidity-controlled room (70 ± 3°F, 35–70% humidity) on a 12 h light/12 h dark cycle (7 a.m. on, 7 p.m. off) and had free access to food and water. All animals younger than postnatal day 10 were allocated into groups based solely on gestational age without respect to sex (both males and females were included). For studies involving rodents older than 10 days, sex was treated as a variable.

### 2.2. Serum and CSF Collection

CSF was collected from the cisterna magna of adult (≥3 months old) or embryonic (E14.5) wild-type CD-1 mice. Independent samples (n) were defined as independent animals for adult samples, and as a CSF pool from all embryos in a single litter for embryonic CSF samples. Samples were maintained on ice, then spun 1000× *g* for 10 min at 4 °C. The supernatant was collected and used for analysis. Blood was collected from cardiac puncture, allowed to clot for 10 min at room temperature, and spun at 500× *g* for 10 min at room temperature. The supernatant serum was collected and used for analysis.

### 2.3. Sample Preparation for LC-MS Analysis of Thyroid Hormone Metabolites from CSF and Serum

Per condition, 5–10 μL of CSF or serum were extracted in 4:6:3 chloroform/methanol/water mixture supplemented with isotopically labeled T3 and T4 (at 100 nM, Cambridge Isotope Laboratories, Tewksbury, MA, USA; CLM-7185-C and CLM-8931-PK), isotopically labeled 17 amino acids (at 1/5000, Cambridge Isotope Laboratories, Tewksbury, MA, USA; MSK-A2-1.2), and isotopically labeled reduced glutathione (at 1 µm, Cambridge Isotope Laboratories, Tewksbury, MA, USA; CNLM-6245-10). After centrifugation for 10 min at maximum speed on a benchtop centrifuge (Eppendorf, Hamburg, Germany), the hydrophilic top layer was transferred to a new tube, dried using a nitrogen dryer (Thermo Fisher Scientific, Waltham, MA, USA; TS-18826), and reconstituted in 20 µL 70% acetonitrile (supplemented with QReSS, Cambridge Isotope Laboratories, Tewksbury, MA, USA; MSK-QRESS-KIT) by brief vortexing. The extracted metabolites were spun again and the cleared supernatant was transferred to LC-MS micro vials. The protocol was used for both fresh and frozen CSF and serum samples. A small amount of each sample was also pooled and serially diluted 3- and 10-fold to be used as quality controls throughout the run of each batch. Serum and CSF sample pools were kept separate and serum and CSF sample sets were run consecutively on our chromatography to avoid interspersing the run of two different matrixes.

### 2.4. Chromatographic Conditions for Polar Metabolite Detection

ZIC-pHILIC chromatography was used to detect polar metabolites in the following manner: 1–2 µL of reconstituted sample was injected into a ZIC-pHILIC 150 × 2.1 mm (5 µm particle size) column (EMD Millipore) operated on a Vanquish™ Flex UHPLC System (Thermo Fisher Scientific, San Jose, CA, USA). Chromatographic separation was achieved using the following conditions: buffer A was acetonitrile; buffer B was 20 mM ammonium carbonate, 0.1% ammonium hydroxide. Gradient conditions were as follows: linear gradient from 20% to 80% B; 20–20.5 min: from 80% to 20% B; 20.5–28 min: hold at 20% B. The column oven and autosampler tray were held at 25 °C and 4 °C, respectively. Chromatographic performance was quality controlled using a mixture of unlabeled standard amino acids and a mixture of chemically diverse compounds (Cambridge Isotope Laboratories, Tewksbury, MA, USA; MSK-A2-US-1.2 and MSK-QRESS-US-KIT) with 1 µL of each injected before or after every run on our HILIC method.

### 2.5. Chromatographic Conditions for T3/T4 Detection with Standards and Biofluids

ZIC-pHILIC chromatography was performed as described above for polar metabolites. A total of 1 µL of reconstituted standards was injected.

Accucore-HILIC chromatography: 1 μL of reconstituted standards was injected into a Thermo Fisher Scientific™ Accucore™ 150 Amide HILIC (150 × 3 mm, 2.6 mm particle size; Thermo Fisher Scientific Waltham, MA, USA). Chromatographic separation was achieved using the following conditions: buffer A was acetonitrile; buffer B was 20 mM ammonium carbonate, 0.1% ammonium hydroxide. Gradient conditions were as follows: 0–20 min: linear gradient from 20% to 80% B; 20–20.5 min: from 80% to 20% B; 20.5–28 min: hold at 20% B. The column oven and autosampler tray were held at 35 °C and 4 °C, respectively.

LUNA-NH_2_ chromatography: 1 μL of reconstituted standards was injected into a Luna^®^ 3 µm NH_2_ 100 Å, LC Column (150 × 2 mm, 3 µm particle size; Phenomenex, 00F-4377-B0). Chromatographic separation was achieved using the following conditions: buffer A was acetonitrile; buffer B was 5 mM ammonium acetate and 0.2% ammonium hydroxide. Gradient conditions were as follows: 20 min linear gradient from 10% to 90% B; 20–25 min hold at 90% B; 25–26 min from 90% to 10% B; 26–34 min hold at 10% B. The column oven and autosampler tray were held at 30 °C and 4 °C, respectively.

C18 chromatography: 1 μL of reconstituted standards or 5–7 µL of reconstituted sample was injected onto an Ascentis Express C18 HPLC column (2.7 μm × 15 cm × 2.1 mm; Sigma Aldrich, Burlington, MA, USA). The column oven and autosampler tray were held at 30 °C and 4 °C, respectively. The following conditions were used to achieve chromatographic separation: buffer A was 0.1% formic acid; buffer B was acetonitrile with 0.1% formic acid. The chromatographic gradient was run at a flow rate of 0.250 mL min^−1^ as follows: 0–5 min: gradient was held at 5% B; 2–12.1 min: linear gradient of 5% to 95% B; 12.1–17.0 min: 95% B; 17.1–21.0 min: gradient was returned to 5% B.

### 2.6. MS Data Acquisition Conditions for Targeted Analysis of Polar Metabolites and Thyroid Hormones

MS data acquisition was performed using a Q Exactive Orbitrap benchtop orbitrap mass spectrometer equipped with an Ion Max source and a HESI II probe (Thermo Fisher Scientific, San Jose, CA, USA) in positive and negative ionization mode in a range of *m*/*z* = 70–1000, with the resolution set at 70,000, the AGC target at 1 × 10^6^, and the maximum injection time (Max IT) at 20 msec. A narrower scan in positive mode at *m*/*z* = 600–800 was used for more specific detection of TH. The resolution was set at 70,000, the AGC target was 5 × 10^5^, and the max IT was 100 msec. For polar metabolites, HESI conditions were as follows: sheath gas flow rate: 35 units; Aug gas flow rate: 8 units; sweet gas flow rate: 1 unit; spray voltage: 3.5 kV (pos), 2.8 kV (neg); capillary temperature: 320 °C; S-lens RF: 50; Aux gas heater temperature: 350 °C. For T3/T4, HESI conditions were as follows: sheath gas flow rate: 40 units; Aug gas flow rate: 10 units; sweet gas flow rate: 0; spray voltage: 3.5 kV (pos), 2.8 kV (neg); capillary temperature: 380 °C; S-lens RF: 60; Aux gas heater temperature: 420 °C.

### 2.7. Targeted Metabolomics Data Analysis

Relative quantification of polar metabolites was performed with TraceFinder 5.1 (Thermo Fisher Scientific, Waltham, MA, USA) using a 5 ppm mass tolerance and referencing an in-house library of chemical standards (see associated Appendix A). We routinely queried 266 compounds (40 internal standards and 226 metabolites). Pooled samples and fractional dilutions were prepared as quality controls and injected at the beginning and end of each run. In addition, pooled samples were interspersed throughout the run to control for technical drift in signal quality as well as to serve to assess the coefficient of variability (CV) for each metabolite. Data from TraceFinder were further consolidated and normalized with an in-house R script, freely accessible at github (https://github.com/FrozenGas/KanarekLabTraceFinderRScripts/blob/main/MS_data_script_v2.4_20221018.R). Briefly, this script performs the following normalization and quality control steps: (1) extracts and combines the peak areas from TraceFinder output.csvs; (2) calculates and normalizes to an averaged factor from all mean-centered chromatographic peak areas of isotopically labeled amino acid and QReSS internal standards within each sample; (3) filters out low-quality metabolites based on user-inputted cut-offs calculated from pool reinjections and pool dilutions; (4) calculates and normalizes for biological material amounts based on the total integrated peak area values of high-confidence metabolites. In this study, the linear correlation between the dilution factor and the peak area cut-offs is set to RSQ > 0.95 and the coefficient of variation (CV) < 30%. Finally, data were log transformed and Pareto scaled within the MetaboAnalyst-based statistical analysis platform [42] to generate PCA, PLSDA, volcano plots, and heatmaps. Individual metabolite bar plots and statistics were calculated in Excel (v16.81) and GraphPad Prism (v.10).

### 2.8. Data Analysis for TH Levels

Data analysis for T3 and T4 levels was performed with TraceFinder 5.1 as described above, referencing empirically determined retention times using standards. Isotopically labeled T3 and T4 internal standards (Cambridge Isotope Laboratories, Tewksbury, MA, USA; CLM-7185-C and CLM-8931-PK) were used to normalize for technical signal variability and matrix effects. Equal volumes of samples were compared per condition.

## 3. Results

### 3.1. Optimization of LC-MS Conditions for the Detection of T3 and T4 Hormones in Mouse CSF

To establish a robust protocol to measure T3 and T4 from mouse CSF by LC-MS, we optimized both chromatographic and mass spectrometry conditions. To do this, we employed synthetic standards and first tested different heated electrospray ionization (HESI) conditions on a high-resolution mass spectrometer (Q Exactive Orbitrap, Thermo Fisher Scientific, New York, NY, USA). We sequentially varied the S-lens values or the capillary and auxiliary gas temperatures on independent chromatographic gradient separations (for further details on chromatography, see below). The corresponding peaks were integrated, manually inspected, and total peak areas were compared (Figure 1A). For capillary and auxiliary (Aux) gas temperatures, we generally kept the Aux gas ~50 °C higher than the capillary (Cap) temperature. The exact conditions tested are given in Appendix A. For both compounds, we observed clear optimal parameters, which we then maintained for subsequent experiments. These were as follows: S-lens = 70 and Aux/Cap Temp = 400/450 °C.

As the next step, we tested different chromatographic conditions. We explored both different polar (HILIC) and non-polar methods (reverse phase). We observed various degrees of peak separation and quality (Figure 1B,C) depending on the mobile and stationary phase combination. The sharpest peaks and highest ion abundances were achieved using reverse-phase (Figure 1C) chromatography. Among the HILIC conditions, the best results were obtained with the Accucore Amide column; however, we could not achieve peak separation between T3 and T4 hormones using our standard conditions for this column. We determined the linear dynamic range and limit of detection (LOD) for each of the tested chromatographic methods. The LODs for T3 and T4 were overall higher using HILIC chromatography (ZIC-pHILIC at 3.8 and 1.5 nM; LUNA NH_2_ at 47.6 and 15.2 nM and Accucore Amide at 0.1 nM for both hormone forms) (Appendix A). The LODs for T3 and T4 were substantially lower using reverse-phase chromatography, at 0.4 and 0.5 nM, respectively (Figure 1D). Thus, we chose to use reverse-phase chromatography for subsequent experiments.

As a final characterization step of the synthetic T3 and T4 standards, we obtained MS^2^ spectra at different collision energies, which serve as compound annotation and verification when using biological matrices (Figure 1E, Appendix A). We publicly deposited these spectra in MetabolomicsWorkbench as a resource for the community.

### 3.2. Optimization of CSF Sample Preparation for T3 and T4 Detection by LC-MS

Our next goal was to determine the optimal extraction and reconstitution conditions and volumes for the robust detection of T3 and T4 from mouse CSF. We first tested two different extraction conditions: either an 80% methanol extraction or a two-phase extraction with chloroform/methanol/water at a ratio of 4:6:3. To compare these two extraction conditions, CSF was collected from three independent mice and each sample was split in half and extracted both ways. In two-phase extraction, T3 and T4 both partitioned in the top, “polar” phase, and signal intensity and variability were superior in the polar phase compared to both the bottom, “non-polar“ phase and to extraction in methanol alone (Figure 2A). This could represent T3 and T4 being more effectively released from protein partners during extraction in a more aggressive solvent such as chloroform. We conclude that a two-phase extraction is better suited than methanol alone.

The analysis of mouse CSF is limited by the relatively small volumes available per replicate (n) (3–15 µL [21,43]), which may need to be further aliquoted across different assays. This limitation of small volume can be partially mitigated by pooling samples into a single replicate (N), which has the additional benefit of mitigating inter-individual variability. Nevertheless, mouse CSF remains a limited sample. Thus, to optimize sample usage, we tested the lower limits of detection by investigating signal intensity and reproducibility in two different volumes of CSF: 1 µL and 5 µL. We observed correspondingly lower signals with a smaller amount of input material, indicating that detection occurred in a linear range with no significant ion suppression with either initial CSF volume input (Figure 2B). However, the signal from 1 μL CSF was below our estimated LOD (obtained from standards). Therefore, we conducted subsequent measurements with volumes larger than 1 μL, and with >5 μL whenever possible.

To further minimize the effects of interfering ions during detection and to obtain a better signal-to-noise ratio, we tested the effects of different mass spectrometer scanning parameters. In a tSIM experiment, the isolation window for the quadrupole mass analyzer was set to 1 *m*/*z* and centered on the individual *m*/*z* values for T3 or T4 (651.7973 and 777.6940, respectively). In a “narrow” experiment, the quadrupole mass analyzer was operated in a full scan mode but set to a narrow range (600–800 *m*/*z*) to capture both T3 and T4 and exclude smaller or larger ions. We did not observe a significant difference between these two conditions (Figure 2B) and chose to continue with the narrow scan for future experiments, as it provides more scans over each peak (one scan per cycle is required in a narrow experiment vs. two scans in a tSIM experiment) and thus improves quantitation.

Many small molecules extracted from biological materials are difficult to reconstitute after extraction solvents evaporate [44,45]. Since our starting material was very limited, it was crucial to maximize reconstitution and concentrate the extracted metabolites after the drying step. We tested two different solvents to resolubilize the metabolites: water and 70% ACN (30% water). To exclude any variability introduced by the natural abundance of T3 and T4 in CSF samples, we performed this solubilization test using CSF with ^13^C-T3 and ^13^C-T4 spiked into the extraction buffer. We included CSF rather than just using the extraction buffer on its own to control for any effect the matrix itself could exert on solubility. We obtained slightly better results with the ACN reconstitution, as observed by plotting the relative abundance of each internal standard after CSF extraction and reconstitution (Figure 2C and Appendix A; see also Appendix A). We chose to continue future experiments with the 70% ACN reconstitution. We also noted that a significant amount of variability persisted between each replicate, even for the spiked ^13^C-labeled standards. These results indicated the importance of including internal standards for proper normalization to correct for per-sample solubility differences and other technical variabilities.

Finally, we validated these combined optimum parameters by comparing the relative abundance of T3 and T4 in rat CSF and serum. We observed significantly higher levels of both hormones in serum compared to CSF (Figure 2D), recapitulating previously published results [21,37] and thus confirming the suitability of our approach. It is important to acknowledge that we did not conduct equivalent tests for the extraction efficiency of T3 and T4 from serum. However, we believe that the methodology outlined here is appropriately robust for this purpose.

### 3.3. Investigation of the Effect of Handling and Storage Conditions on T3 and T4 Quantitation from Mouse CSF

Obtaining sufficient quantities of mouse CSF can be challenging and time-consuming and common experimental conditions can require sample collections days apart; therefore, fresh samples are not always available for analysis. This is particularly important if using this method to analyze TH in human clinical CSF samples, which are frozen. Thus, we determined the impact of freezing/thawing and storage conditions on T3 and T3 measurements using the newly optimized LC-MS method. We compared the relative abundance of T3 and T4 extracted from the same CSF samples, which were either fresh, acutely frozen and then thawed (labeled “frozen”), or frozen and then stored at −80 °C for 24 h (labeled “stored”) before thawing and extraction (Figure 3A). Reassuringly, a single freeze–thaw cycle or short-term storage did not affect total abundances. We repeated this experiment including a larger cohort of adult mice, which included serum and mock extraction samples (Figure 3B). As observed previously, we detected a robust signal from mouse serum and no difference in signal between fresh and frozen CSF samples (see also Appendix A). Of note, we recommend the inclusion of mock samples when analyzing T3 and T4 because system signal carryover or contamination (perhaps due to the poor solubility of T3 and T4) can exist and actual levels can be quite close to the LODs and thus hard to distinguish from the background without these crucial controls. We conclude that when larger experiments are required and samples cannot all be obtained on the same day, the storage of frozen CSF can be an acceptable alternative. We cannot extrapolate these results to long-term storage or multiple cycles of freezing and thawing. Nevertheless, for subsequent experiments, we measured fresh samples whenever possible.

### 3.4. Comparison of T3 and T4 Levels in Adult and Embryonic CSF

To further validate our T3 and T4 detection method and harness its utility to make discoveries about CSF TH, we interrogated the levels of these hormones in the setting of developmental changes in CSF. To our knowledge, mouse embryonic CSF TH have not been rigorously investigated and little is known about the levels of these hormones relative to adult mouse CSF and serum. We analyzed T3 and T4 in embryonic stage E14.5 and in adult (male and female) CSF and in serum from the same adult male mice. Interestingly, we observed significantly higher levels of T3 in embryonic CSF relative to adult CSF or serum (Figure 4A and Appendix A). T4 levels in embryonic CSF were also slightly elevated compared to those in adult CSF, but the difference was smaller and more variable between experiments (Figure 4A,B). This intra- and inter-experimental variability could, in part, stem from metabolic or circadian changes between the animals due to small changes in collection times [21]. Notwithstanding this, T4 levels in embryonic CSF were below adult serum levels, in contrast to what we observed for T3. Further, the levels of each hormone in frozen samples were similar to those found in fresh samples for each experiment (Appendix A), confirming our previous observation that short-term storage does not significantly impact the relative quantification of T3 and T4. We further compared the levels of T3 and T4 in adult male and female mice (Figure 4A and Appendix A). T3 and T4 levels were slightly reduced in females compared to males. Strikingly, pregnant dams had significantly higher levels of CSF T4 than either non-pregnant adults or embryos (Figure 4B). This aligns with previous reports that TH levels increase during pregnancy and further validates our method [46]. Higher TH levels in pregnant dams suggest an etiology for the higher levels of T3 and T4 in embryonic CSF, which are largely formed maternally and transferred to embryos through the placenta [11].

We also separately optimized sample preparation conditions for embryonic samples as they could be different relative to adults, given the higher TH concentrations and potential differences in matrix effects. Consistently, the best extraction was achieved with the two-phase method (Appendix A), as observed before. Further, as we relied on internal standards for normalization, we are confident that any potential differences in matrix effects between the different biofluids were effectively accounted for.

Finally, to maximize discovery from the limited sample sets, we further employed targeted metabolomics profiling of a panel of around 200 compounds using these samples. We interrogated relative changes between fresh and frozen adult (male or female) CSF compared to fresh and frozen embryonic CSF (Figure 4C,D and Appendix A). Embryonic CSF samples were quite distinct from adult CSF as indicated by PCA analysis (Figure 4C,D), while much smaller differences were observed between fresh and frozen CSF samples (Appendix A) or between CSF from males and females (Appendix A). Nevertheless, significant differences emerged between CSF from male vs. female adults (Appendix A) and between fresh vs. frozen embryonic CSF (Appendix A). Notably, many of the metabolites that differed between male and female CSF were amino acids. In contrast, metabolites that were different between fresh and frozen embryo samples were related to de novo and salvage nucleotide biosynthesis (guanosine, cytosine, adenosine, inosine, UMP, CMP, guanine, cytidine, uridine, hypoxanthine) or oxidative stress (GSSG, homocysteine sulfinic acid). In light of these differences between fresh and frozen samples, we recommend including freezing and storage as variables in experiments with labile CSF metabolites and minimizing their effects whenever possible. These broad metabolomics data revealed important new differences in adult CSF by sex and between adult and embryonic CSF. They also establish a reference for future experiments and inform the broader community on the levels of metabolites in CSF—a biofluid that is particularly difficult to obtain in sufficient amounts for metabolite profiling.

## 4. Discussion

Here, we report an LC-MS-based method to quantitate TH (T3 and T4) and polar metabolites in embryonic and adult CSF and in adult serum. We tested several chromatographic conditions, reported their respective linear limits of detection (LODs), and found reverse-phase chromatography to be the most suitable. We further optimized HESI ionization parameters for sensitive detection of T3 and T4 on an Orbitrap mass spectrometer. In parallel, we explored several conditions for sample preparation, reporting optimal recovery with a two-phase extraction based on chloroform, methanol, and water. We further determined that a freeze–thaw cycle and/or short-term storage of CSF and serum samples do not significantly affect T3 and T4 signal intensity and are thus acceptable when samples cannot be analyzed on the same day of an experiment. Finally, a significant advantage of the method described here is that it can be combined with parallel analysis of central energy metabolism. This allows for comprehensive profiling of T3 and T4 levels with other metabolic pathways and may reveal correlative and associated metabolic effects.

Our method reliably detects both T3 and T4 levels in plasma and adult and embryonic CSF. However, we noted that T3 levels were sometimes quite close to our LODs. It is beyond the scope of this study to explore further chromatographic columns and conditions. Although we achieved the best sensitivity with reverse-phase chromatography, we noted that the Accucore Amide column also performed well, despite not being able to separate T3 from T4. These results are consistent with previous studies that also primarily relied on reverse-phase chromatography [32,33,34,35]. Therefore, further exploration of chromatographic conditions and columns could lead to improved sensitivity and linear dynamic ranges. As an orthogonal approach to improved signal-to-noise ratio, CSF could be pooled from individual mice to achieve larger quantities and therefore increased recovery. Thus, further optimization of sample volumes, perhaps from less limiting sources, such as rat or human CSF, could be explored in the future.

The primary goal of this study was to assess the total levels of bioavailable T3 and T4 hormones in biofluids; thus, we did not differentiate between free or transthyretin-bound TH. The majority of TH in CSF are bound to carrier proteins, and in serum, 0.03% of total serum T4 and 0.3% of total serum T3 are present in the free or unbound form [47,48,49]. Measuring total T3 and T4 is a reliable index of clinical thyroid status in the absence of protein-binding abnormalities. Notably, methods that preferentially detect these free forms using LC-MS technology rely on equilibrium dialysis or ultrafiltration techniques that would require larger sample volumes than those available from mouse CSF [30]. Free T3 and T4 levels in plasma have been reported in the range of 10–30 pg/mL and 3–6 pg/mL for free T4 and free T3, respectively [50,51]. While we believe that assessing free TH in CSF could be achievable using LC-MS technology, this was beyond the scope of the current manuscript.

TH are critical to embryonic brain development; therefore, our method to monitor CSF TH will help to determine their roles. Due to the dynamic processes that occur sequentially during brain development, the actions of TH during different days of brain maturation will influence many developmental processes and stages, usually during limited time windows. Fetal TH depend on both transplacental passage of maternal TH [52] and the onset of fetal thyroid function (around 18 weeks post gestation in humans/ E17 in rodents) [53]. Maternal hypothyroidism is the most common cause of fetal TH deficiency, but the deficit may also arise in the fetus. In cases of congenital hypothyroidism, babies are born with an underactive or absent thyroid gland. If this is accompanied by normal TH placental transfer, fetal levels can be close to normal at birth then drop quickly postnatally. Because TH play such important roles in brain development and growth, TH screening is often standard near birth and early detection and treatment of hypothyroidism generally result in normal growth and development. However, once TH are in the bloodstream, they must actively traverse brain barriers to access and support the developing brain. These barriers include the blood–brain barrier and the blood–CSF barrier of the choroid plexus, which harbor suites of TH transporters. Defects in these transporters, including MCT8 [13,54], are linked to neurodevelopmental disorders, indicating that not only blood TH levels but also CNS levels are key to monitoring TH effects on brain development. In adults with mild hypothyroidism, lower CSF TH are associated with higher rates of depression and worse quality of life [55]. These results raise the possibility that other neurodevelopmental disorders may also be linked to underlying CNS TH disruptions that may be largely missed by the current blood-based screening methods. Therefore, the new methods to detect CSF TH levels we present, due to their high reproducibility and sensitivity, have the potential to indicate other disruptions in brain development that result from deficient TH in the CNS even when peripheral measurements indicate euthyroid conditions.

## 5. Conclusions

This study presents a reliable LC-MS method for quantitatively assessing T3 and T4 and polar metabolites in CSF and serum samples. The optimized chromatographic conditions and sample preparation techniques described here will serve the community by enabling TH detection in fresh or frozen CSF samples, including limited samples such as those collected from mouse embryos. While the method successfully detects TH in CSF, there is room for further improvement by exploring different chromatographic conditions to enhance sensitivity, as well as investigating free TH levels in CSF.

## Figures and Tables

**Figure 1 metabolites-14-00079-f001:**
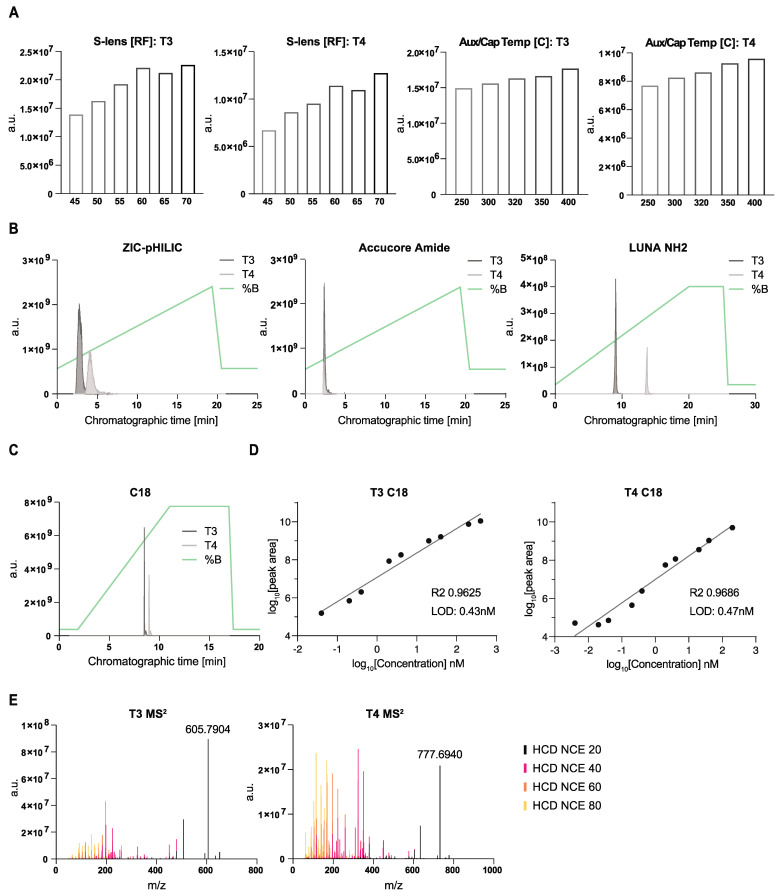
A method to quantify triiodothyronine (T3) and thyroxin (T4) by LC-MS. (**A**) Optimization of HESI parameters on Orbitrap mass spectrometer for the indicated metabolites using C18 chromatography and standards. Graphs represent integrated areas of chromatographic peaks under changing parameters for capillary temperature or S-lens. (**B**,**C**) Retention times and sensitivity comparison among four chromatography methods. Chromatographic runs were carried out on either “ZIC-pHILIC” (ZIC-pHILIC 150 × 2.1 mm, 5 µm particle size, EMD Millipore); “Accucore Amide” (Accucore™ 150 Amide HILIC (150 × 3 mm, 2.6 mm particle size; Thermo Fisher Scientific)); “LUNA NH_2_” (Luna^®^ 3 µm NH_2_ 100 Å, LC Column (150 × 2 mm, 3 µm particle size; Phenomenex, 00F-4377-B0)); or “C18” (Ascentis Express C18 HPLC column (2.7 µm × 15 µm × 2.1 mm; Sigma Aldrich)) with 22 min (polar chromatography) or 18 min (C18 reverse-phase chromatography) linear gradients, respectively. Overlaid peaks are shown for the indicated ranges of (**B**) polar and (**C**) reverse-phase chromatographic methods. Overlayed are corresponding chromatographic gradient compositions. Metabolite stock solutions were diluted in 10 µm ammonium hydroxide solution at 1:1 ratio of methanol/water. (**D**) Limit of detection (LOD) and linearity for individual T3 and T4 standards, respectively, on C18 reverse-phase chromatography. Presented is one experiment from two representative dilutions; R^2^—goodness of fit; “LOD”—linear limit of detection. (**E**) MS^2^ spectra comparison to different collisions for T3 and T4 standards.

**Figure 2 metabolites-14-00079-f002:**
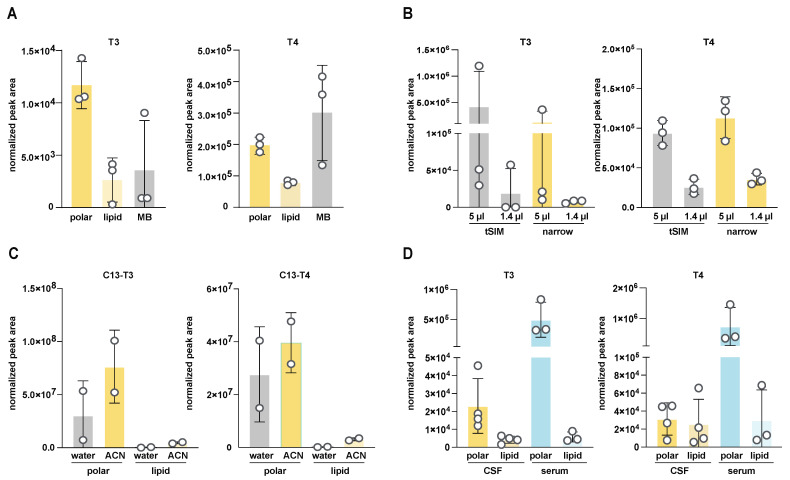
Optimization of extraction and reconstitution conditions for T4 and T3 from cerebrospinal fluid (CSF). (**A**) Optimization of solvents for extraction of T3 and T4 from CSF. Single-phase (“MB”: methanol-based; see methods for further details) vs. two-phase (lipid and polar phase, respectively) extractions are compared. Normalized peak integration areas are shown as the average and standard deviation for CSF collections in triplicate. (**B**) Fine-tuning of mass spectrometry detection parameters. Two narrower detection scans (tSIM vs. “narrow”, corresponding to 600–800 *m*/*z*) were compared for T3 and T4 extracted from either 5 μL or 1.4 μL of CSF. Normalized peak integration areas are shown as the average and standard deviation for CSF samples in triplicate. (**C**) Comparison of reconstitution efficiency for isotopically labeled T3 and T4 standards spiked in mouse CSF (see also Appendix A). Both lipid (bottom) and polar (top) phases from two-phase extraction were compared, where each was reconstituted in either water or 70% ACN. Normalized peak integration areas are shown as the average and standard deviation for two independent extractions. (**D**) Comparison of the partitioning efficiency of extracted T3 or T4 between the polar and lipid phase of a two-phase extraction for rat CSF or serum. Normalized peak integration areas are shown as the average and standard deviation for quadruplicate CSF collection and triplicate serum collection.

**Figure 3 metabolites-14-00079-f003:**
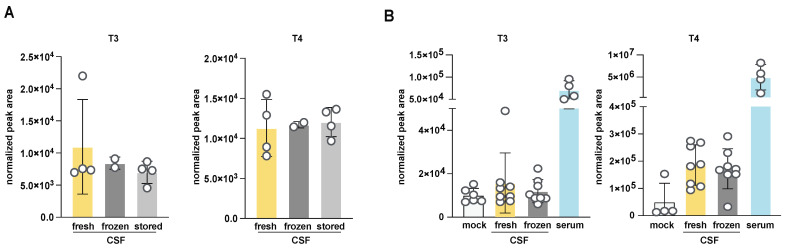
Characterization of T3 and T4 extracted from CSF and serum after different storage conditions. (**A**) Comparison of storage conditions and signal stability upon freezing and storage for rat CSF and serum T3 or T4. Freshly extracted CSF extract was compared to fresh, acutely frozen and then thawed (labeled “frozen”), and frozen and then stored at −80 °C for 24 h (labeled “stored”). Bottom and top phases from a two-phase extraction are further compared. Normalized peak integration areas are shown as the average and standard deviation for at least a triplicate CSF collection. (**B**) As in (**A**) but mouse serum levels of T3 and T4 were compared to two storage conditions of mouse CSF and a mock sample (empty tube processed as samples). CSF was collected from at least 8 mice with paired serum from 4 mice.

**Figure 4 metabolites-14-00079-f004:**
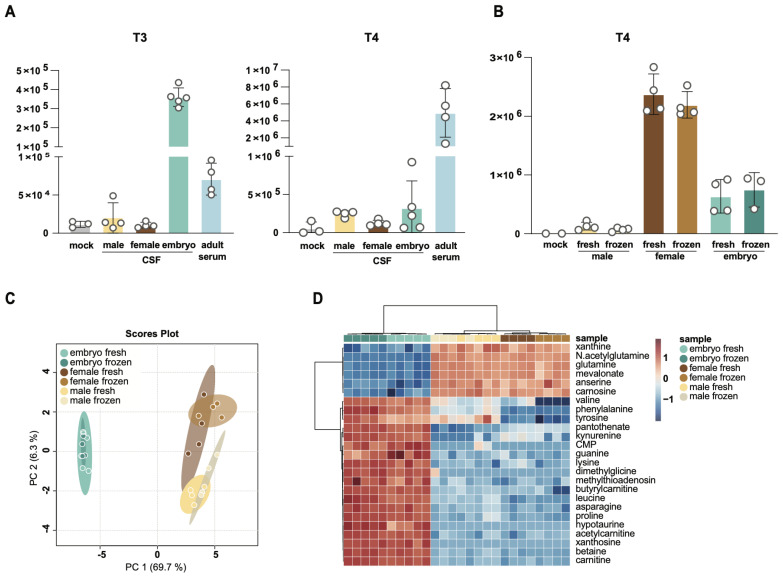
Levels of T3 and T4 differ between adult, embryo, and pregnant mice. (**A**) Comparison of T3 and T4 levels extracted from adult and embryonic mouse CSF and adult serum. Normalized peak integration areas are shown as the average and standard deviation for at least five CSF collections and four paired serum collections. (**B**) Comparison of T4 levels extracted from CSF of adult males, pregnant females, or embryos. Normalized peak integration areas are shown as the average and standard deviation for at least eight CSF collections. Missing values (zeros) were excluded from the analysis. (**C**,**D**) PCA (**C**) and heatmap (**D**) analyses of embryo, female, and male CSF polar metabolites using HILIC chromatography LC-MS. Detected metabolites were Pareto scaled and log transformed within the MetaboAnalyst online platform. (**D**) Top 25 changed metabolites are shown.

## Data Availability

Data associated with this study will be made available at MetabolomicsWorkbench: https://www.metabolomicsworkbench.org/.

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
