# Peer review of "Optimized Mass Spectrometry Detection of Thyroid Hormones and Polar Metabolites in Rodent Cerebrospinal Fluid"

_metabolites, 2024, doi:10.3390/metabo14020079_

Round 1

Reviewer 1 Report

Comments and Suggestions for Authors

The article "Optimized Mass Spectrometry Detection of Thyroid Hormones and Polar Metabolites in Rodent Cerebrospinal Fluid" shows the development of the detection method. However, in some places it is difficult to understand what the author wants to say and what he wants to focus on.

Author Response

Author’s response

Reviewer 1: The article "Optimized Mass Spectrometry Detection of Thyroid Hormones and Polar Metabolites in Rodent Cerebrospinal Fluid" shows the development of the detection method. However, in some places it is difficult to understand what the author wants to say and what he wants to focus on.

We thank the reviewer for the positive comments and have edited the entire manuscript again to improve clarity and enhance the focus of each section. We attach a marked version of the manuscript with highlighted changes. Please see below all relevant significant changes to our manuscript.

Abstract:

Line16-18: We report an optimized liquid chromatography-mass spectrometry (LC-MS) based method to measure TH in rodent CSF and serum, applicable to both fresh and frozen samples. Using this new method, we find distinct differences in CSF thyroid hormone in pregnant dams vs.

Introduction:

Line 34-37: In sum, maintaining balanced TH levels within the central nervous system (CNS) is vital for developmental health. Therefore, measuring TH concentrations directly within the CNS, rather than in the peripheral system, could enhance the accuracy of monitoring these essential biological pathways.

Line 44-50: The choroid plexus tissues are 3-dimensional structures of specialized cuboidal epithelial cells surrounding fenestrated capillaries and mesenchyme that extend into the four brain ventricles. The choroid plexus secretes CSF in adults and comprises the blood-CSF barrier by limiting transcellular transport and maintaining tight junctions at the apical surface of the choroid plexus epithelial cells. Choroid plexus epithelial cells transport TH into the CSF by TH carrier protein transthyretin (TTR), which is the major transporter of TH hormone, and through transporters MCT8 and OAT1C1[16]–[18].

Line 74-79: In the past, radioimmunoassays were the standard for measuring TH levels, but these methods were constrained by their narrow sensitivity and dynamic range. Liquid chromatography-mass spectrometry (LC-MS) techniques offer a substantial enhancement, providing a broader dynamic range and heightened sensitivity. Such sensitivity is especially useful for TH detection as they are typically found in trace amounts, ranging from 1.4 to 2.3 pg/mL in biological samples[36], and their levels fluctuate dynamically.

Results:

Line 322-327: Since our starting material was very limited, it was crucial to maximize reconstitution and concentrate extracted metabolites after the drying step. We tested two different solvents to re-solubilize metabolites: water and 70% ACN (30% water). To exclude any variability introduced from the natural abundance of T3 and T4 in CSF samples, we performed this solubilization test using CSF with 13C-T3 and 13C-T4 spiked into the extraction buffer.

Line 340-343: It's important to acknowledge that we have not conducted equivalent tests for the extraction efficiency of T3 and T4 from serum. However, we believe that the methodology outlined here is appropriately robust for this purpose.

Line 394-396: Freshly extracted CSF extract was compared to fresh, acutely frozen and then thawed (labeled “frozen"), or frozen and then stored at -80 °C for 24 h (labeled “stored”).

Line 430-432: Further, as we rely on internal standards for normalization, we are confident that any potential differences in matrix effects between the different biofluids are effectively accounted for.

Discussion:

Line 513-514: TH is critical to embryonic brain development and therefore this method to monitor CSF TH will inform these roles.

All changes to our manuscript are further highlighted in the attached marked version of the manuscript

Reviewer 2 Report

Comments and Suggestions for Authors

Manuscript “Optimized Mass Spectrometry Detection of Thyroid Hormones and Polar Metabolites in Rodent Cerebrospinal Fluid” by Fame et al., described that the  TH detection and metabolite profiling of related metabolic pathways open new avenues of rigorous research into CSF thyroid hormone and will inform future studies on metabolic alterations in CSF during normal development  by using an optimized LC-MS based method to measure thyroid hormones in rodent CSF and serum, applicable to both fresh and frozen samples. They found the distinct differences in CSF thyroid hormone in pregnant dams vs. non-pregnant adults and in embryonic vs. adult CSF. Further, targeted LC-MS metabolic profiling uncovers distinct central carbon metabolism in the CSF. Authors advocate that this method successfully detects TH from CSF, and further improvement can be possible by exploring different chromatographic conditions to enhance sensitivity, as well as investigating free TH levels in CSF. Thus the present work contributes significantly towards the improvement of detection methods for TH in serum and CSF as well as fresh or frozen samples, further optimization of sample volumes, perhaps from less limiting sources such as rat or human CSF, could be explored in the future. The manuscript described and utilized the advanced detection methods. The new methods  presented to detect CSF TH levels with high reproducibility and sensitivity have the potential to inform other disruptions in brain development that result from deficient TH in the CNS even when peripheral measurements indicate euthyroid conditions.

I found the research work presented is of quite significance and methodology used is advanced, results obtained were well explained to elucidate the mechanisms involved. I have no hesitation to consider it for favor of publication in its present form except any editorial changes if any.

Author Response

We thank the reviewer for this precise and elegant description of our work and for sharing our enthusiasm for the potential of this new approach to boost study of biofluid metabolites and TH.

We have revised our manuscript to improve readability and have introduced minor changes to clarify methodology and results. All changes to our manuscript are further highlighted in the attached marked version of the manuscript.

Reviewer 3 Report

Comments and Suggestions for Authors

In the paper named “Optimized Mass Spectrometry Detection of Thyroid Hormones and Polar Metabolites in Rodent Cerebrospinal Fluid” authors present and interesting more sensitive and quantitative LC-MS method to detect T3 and T4 hormone levels in CSF from both fresh and frozen samples. This method as described the authors is based on high resolution accurate mass detection via Orbitrap MS and on separation of hormone forms on a reverse phase chromatography. Moreover author optimized several aspects of signal detection, explored diverse chromatographic methods, and improved sample preparation. Therefore using this new optimized LC-MS method author confirmed lower levels of TH in CSF compared to serum and further report dynamic developmental and physiological differences.

Only minor questions are required

1)      Why only indicate in methods that used Adult CD-1 male mice and pregnant CD-1 dams and in figure 4 female samples are included?

2)      In point 2.2 they say that perform a pool form embryonic samples them they do not use the pregnant males however use the embryos? Please clarify?

3)      The point 2.3 is identical for fresh and frozen samples?

4)      All the parameters optimization was made in CSF, but author confirm that the same parameters are needed for serum have author made and independent optimization for Serum? The question is due to the different matrix between serum and CFS that can although the metabolites were extracted in the retention time and other chromatographic parameters.

Author Response

Reviewer 3: In the paper named “Optimized Mass Spectrometry Detection of Thyroid Hormones and Polar Metabolites in Rodent Cerebrospinal Fluid” authors present and interesting more sensitive and quantitative LC-MS method to detect T3 and T4 hormone levels in CSF from both fresh and frozen samples. This method as described the authors is based on high resolution accurate mass detection via Orbitrap MS and on separation of hormone forms on a reverse phase chromatography. Moreover author optimized several aspects of signal detection, explored diverse chromatographic methods, and improved sample preparation. Therefore using this new optimized LC-MS method author confirmed lower levels of TH in CSF compared to serum and further report dynamic developmental and physiological differences.

Only minor questions are required

1)      Why only indicate in methods that used Adult CD-1 male mice and pregnant CD-1 dams and in figure 4 female samples are included?

We thank the reviewer for noticing this discrepancy. We have clarified this point by including Male and Female adult CD-1s and pregnant CD-1 dams in methods section. Please see the relevant highlighted edits to our manuscript below:

Line 107-110: The Boston Children’s Hospital IACUC approved all experiments involving mice in this study. Adult CD-1 male and female mice and pregnant CD-1 dams were obtained from Charles River Laboratories. Sprague Dawley rats were purchased from Charles River Laboratories.

2)      In point 2.2 they say that they perform a pool from embryonic samples but they do not use the pregnant females however use the embryos? Please clarify?

We thank the reviewer for pointing out this ambiguity. We have now added text in the Methods section to clarify what samples were used. Indeed, we pooled embryonic samples, but did not pool adult samples, so a biological replicate for adults is a single individual. There were quality control pools used later as described in 2.3 and 2.6, but these were pooled post extraction, as opposed to the littermate embryonic CSF samples, that were pooled pre-extraction, and that were treated as a single biological sample. We have highlighted below the relevant changes to the manuscript:

Line 118-120: Independent samples (n) were defined as independent animals for adult samples, and as a CSF pool from all embryos in a single litter for embryonic CSF samples.

3)      The point 2.3 is identical for fresh and frozen samples?

Thank you for noting this confusion, we have now clarified that the same procedures were used on fresh and frozen samples. We have highlighted below the relevant changes to the manuscript:

Line 135-140: Extracted metabolites were spun again and cleared supernatant was transferred to LC-MS micro vials. The protocol was used for both fresh and frozen CSF and serum samples. A small amount of each sample was also pooled and serially diluted 3- and 10-fold to be used as quality controls throughout the run of each batch. Serum and CSF sample pools were kept separate and serum and CSF sample sets were run consecutively on our chromatography to avoid interspersing the run of two different matrixes.

4)      All the parameters optimization was made in CSF, but author confirm that the same parameters are needed for serum have author made and independent optimization for Serum? The question is due to the different matrix between serum and CFS that can although the metabolites were extracted in the retention time and other chromatographic parameters.

Thank you for noting this missing information. Indeed, we have not independently optimized extraction for serum samples. However, we include 13C-labeled internal standards with every extraction. This is of tremendous help when comparing different matrixes as we can be sure that retention times and ionization efficiencies are appropriately controlled for. We have now added text where appropriate to clarify this point in the text. Please, see below all relevant changes highlighted.

Line 135-140: Extracted metabolites were spun again and cleared supernatant was transferred to LC-MS micro vials. The protocol was used for both fresh and frozen CSF and serum samples. A small amount of each sample was also pooled and serially diluted 3- and 10-fold to be used as quality controls throughout the run of each batch. Serum and CSF sample pools were kept separate and serum and CSF sample sets were run consecutively on our chromatography to avoid interspersing the run of two different matrixes.

Line 118-121: Isotopically labelled T3 and T4 internal standards (Cambridge Isotope Laboratories, CLM-7185-C and CLM-8931-PK) were used to normalize for technical signal variability and matrix effects. Equal volumes of samples were compared per condition.

Line 340-343: It's important to acknowledge that we have not conducted equivalent tests for the extraction efficiency of T3 and T4 from serum. However, we believe that the methodology outlined here is appropriately robust for this purpose.

Line 430-432: Further, as we rely on internal standards for normalization, we are confident that any potential differences in matrix effects between the different biofluids are effectively accounted for.

All changes to our manuscript are further highlighted in the attached marked version of the manuscript.
